# Thrombotic microangiopathy after kidney transplantation: Analysis of the Brazilian Atypical Hemolytic Uremic Syndrome cohort

Hong Si Nga[1☯], Lilian Monteiro Pereira Palma[2☯], Miguel Ernandes Neto[1☯], Ida Maria Maximina Fernandes-Charpiot[3‡], Valter Duro Garcia[4‡], Roger Kist[4‡], Silvana Maria Carvalho Miranda[5☯], Pedro Augusto Macedo de Souza[5‡], Gerson Marques Pereira, Jr[5‡], Luis Gustavo Modelli de Andrade[1☯]*

1 Department of Internal Medicine—UNESP, Univ Estadual Paulista, Botucatu, Brazil, 2 Departamento de Pediatria/Hospital de Clínicas, Universidade Estadual de Campinas, Campinas, SP, Brazil, 3 Transplant Unit São Jose Rio Preto/FAMERP, Sao Paulo, Brazil, 4 Transplant Unit Santa Casa de Misericórdia de Porto Alegre–ISCMPA, Porto Alegre, Brazil, 5 Santa Casa de Belo Horizonte, Belo Horizonte, Brazil

☯ These authors contributed equally to this work.
‡ These authors also contributed equally to this work
* gustavo.modelli@unesp.br

**Data Availability Statement:** All files are available from the Kaggle database: https://www.kaggle.com/gustavomodelli/brazilian-ahus-cohort.

# Abstract

## Background

Atypical Hemolytic Uremic Syndrome (aHUS) is an ultra-rare disease that potentially leads to kidney graft failure due to ongoing Thrombotic Microangiopathy (TMA). The aim was evaluating the frequency of TMA after kidney transplantation in patients with aHUS in a Brazilian cohort stratified by the use of the specific complement-inhibitor eculizumab.

## Methods

This was a multicenter retrospective cohort study including kidney transplant patients diagnosed with aHUS. We collected data from 118 transplant centers in Brazil concerning aHUS transplanted patients between 01/01/2007 and 12/31/2019. Patients were stratified into three groups: no use of eculizumab (No Eculizumab Group), use of eculizumab for treatment of after transplantation TMA (Therapeutic Group), and use of eculizumab for prophylaxis of aHUS recurrence (Prophylactic Group).

## Results

Thirty-eight patients with aHUS who received kidney transplantation were enrolled in the study. Patients' mean age was 30 years (24–40), and the majority of participants was women (63% of cases). In the No Eculizumab Group (n = 11), there was a 91% graft loss due to the TMA. The hazard ratio of TMA graft loss was 0.07 [0.01–0.55], p = 0.012 in the eculizumab Prophylactic Group and 0.04 [0.00–0.28], p = 0.002 in the eculizumab Therapeutic Group.

**Funding:** The authors received no specific funding for this work.

**Competing interests:** I declared that the authors Modelli de Andrade LG, Palma LM, and Miranda SMC received fees from Alexion pharmaceutical to Travel grants and honoraria for speaking and participation at meetings. This does not alter our adherence to PLOS ONE policies on sharing data and materials.

## Conclusion

The TMA graft loss in the absence of a specific complement-inhibitor was higher among the Brazilian cohort of kidney transplant patients. This finding reinforces the need of eculizumab use for treatment of aHUS kidney transplant patients. Cost optimization analysis and the early access to C5 inhibitors are suggested, especially in low-medium income countries.

## Introduction

The atypical Hemolytic Uremic Syndrome (aHUS) is an ultra-rare disease with an incidence of approximately 0.5 cases per million inhabitants per year, and more than half of patients affected by this disease have an intrinsic or acquired abnormality related to the complement system [1]. The aHUS is a consequence of the dysregulation of the alternative complement pathway due to genetic factors, such as autoantibodies and mutations in the proteins of the complement system in approximately 60% of cases [2]. The use of plasmapheresis is the primary supportive treatment in suspected aHUS, however, almost 70% of cases progress to renal replacement therapy and death within 3 years after diagnosis [3]. Currently, the treatment of choice [2, 4] for aHUS is the administration of eculizumab, which showed lower rates of relapses and better renal function compared with plasmapheresis [5–7]. The eculizumab is a long-acting humanized monoclonal antibody that inhibits the cleavage of C5 into C5a and C5b and, hence it inhibits deployment of the terminal complement system including the formation of membrane attack complex (C5b-9). The blockade of the terminal complement pathway performed by the drug quickly and sustainably reduces the process of uncontrolled C5 activation [8].

In the scenario of kidney transplantation, prophylaxis of Thrombotic Microangiopathy (TMA) recurrence with plasmapheresis has shown unsatisfactory results with a negative impact on graft and patient survival [9, 10]. The prophylaxis of recurrence with eculizumab was considered more effective and this has been recommended as the first line prevention strategy [11, 12]. However, due to the high cost of this medication, especially in middle-income countries, the evaluation of its effectiveness is extremely important. The primary aim of this study was to evaluate the recurrence of TMA after kidney transplantation in patients with aHUS in a cohort of Brazilian patients who were stratified by the use of the specific complement-inhibitor eculizumab.

## Methods

### Population

This was a multicenter retrospective cohort study including kidney transplant recipients diagnosed with aHUS. We contacted the 118 transplant centers in Brazil through the Brazilian Transplant Association (ABTO) and invited them to report data on the aHUS transplanted patients. The study was approved by the Research Ethics Committee of the Faculty of Medicine of Botucatu-UNESP (CAE#: 2,810,751). All data were collected from previously anonymized and de-identified databases. Since it did not involve identifiable private information, a waiver of informed consent was granted.

**Inclusion criteria.** All patients diagnosed with aHUS either as primary disease or after relapsing in a kidney transplantation between January 1st, 2007 and December 31st, 2019. The last follow-up was December 31st, 2020.

**Exclusion criteria.** Patients who did not fulfill the aHUS diagnostic criteria described below.

## Diagnosis of aHUS

The diagnosis of aHUS was performed using the clinical history and laboratory exams compatible with TMA (microangiopathic hemolytic anemia, increased lactate dehydrogenase > 1.5 upper normal limit, thrombocytopenia, and kidney injury), and by excluding drug use, infections, and secondary causes [13, 14]. All cases were submitted to a post-transplant renal biopsy which had to be compatible with TMA, characterized by deposition of platelet-rich fibrin or occluding thrombi in at least one glomerulus and/or renal artery or arteriole [15, 16]. In all cases, a diagnosis of antibody-mediated rejection was ruled out by histology and by a negative result of the anti-donor antibody (DSA). The ADAMTS 13 (disintegrin and metalloproteinase with a thrombospondin type 1 motif) activity assay was done in all cases after the year 2011, and values above 5% excluding severe deficiency, i.e., Thrombotic Thrombocytopenic Purpura. At the time of diagnosis, autoimmune diseases were also ruled out and the viral serology was performed (cytomegalovirus, BK virus, and HIV). The diagnosis of all aHUS was reviewed following the steps suggested by de Andrade et al. [17].

## Genetic analysis

Genetic analysis was performed according to the indication of each center and this was not required for diagnostic confirmation. The most common test was the aHUS panel which comprised the amplification and sequencing of complete regions of genes encoding ADAMTS13, C3, CD46, CFB, CFHR1, CFHR2, CFHR3, CFHR5, CFI, DGKE, PIGA, THBD, and including 10 bases pairs next to exons. The analyses were performed according to the protocol described by Richards et al. [18].

## Donor Specific Antibody (DSA)

Detection of DSA from January 2000 until the end of the study period was conducted for class I in loci A, B, and C, and for class II loci DP, DQ, and DR using the single antigen by the Luminex technique. A positive result was considered when mean fluorescence intensity (MFI) was greater than 1500 pre-transplant and greater than 300 post-transplant. Before January 2000, the panel was performed using the complement-dependent cytotoxicity (CDC) technique without quantification of specific antibodies.

## Immunosuppression

The immunosuppression protocol and choice of induction therapy were defined at each center according to local protocols. The immunosuppression switch after the aHUS diagnosis was managed by each center according to its protocol as part of the aHUS differential diagnosis process.

## Treatment of microangiopathy

For the treatment of TMA, plasmapheresis, plasma infusion, and treatment with specific complement inhibitors (Eculizumab, eculizumab—Soliris®, Alexion Pharmaceuticals, Cheshire, CT, USA) were used according to the availability.

## Administration of eculizumab

Patients were stratified into three groups according to the use of eculizumab:

No eculizumab use: Patients who did not receive any dose of complement inhibitor before or after transplantation.

Therapeutic Group: patients who received eculizumab at the time of diagnosis of post-transplantation TMA, comprising a loading dose of 900 mg per week for a total of 4 weeks, followed by 1200 mg at week 5 and then 1200mg every 15 days.

Prophylactic Group: Patients who received a 900 mg dose on the day of the surgery before organ transplant reperfusion followed by a 900mg dose after 24 hours, and then 1200mg every 15 days.

No patient was discontinued eculizumab after starting treatment.

## Predictor variables

All data was collected through an unidentified form. Demographics and epidemiological data, transplant characteristics and immunosuppression.

Demographics and epidemiological data, i.e, gender, age at diagnosis, ethnicity, panel reactivity antibody (PRA), underlying kidney disease, presence of comorbidities, dialysis method (peritoneal, hemodialysis or preemptive), time on dialysis, living or deceased donor transplantation, number of HLA mismatches, donor's age and cause of deceased donor death.

Immunosuppression data: induction therapy (basiliximab, thymoglobulin or no induction) and maintenance therapy (combination of calcineurin inhibitor with mycophenolate or azathioprine or the combination of calcineurin inhibitor with mTOR inhibitor).

Acute rejection was clinically presumed (elevation of serum creatinine of at least 20% compared to baseline) or was biopsy-proven within the first year of transplantation. Time lapse between aHUS relapse and acute rejection was calculated. All biopsy-proven rejections were described according to Banff 2017 criteria [19].

## TMA recurrence

TMA recurrence was defined as the presence of two or more of the following presentations: 1—thrombi in renal graft biopsy; 2—acute kidney injury (> 50% increase in baseline creatinine); 3—thrombocytopenia (platelet count <150,000/μL); 4 –microangiopathic hemolytic anemia (Hb <10 g/dL, lactate dehydrogenase more than 1.5 times reference upper limit value, consumed haptoglobin, presence of schistocytes in the peripheral blood).

## Groups

Three groups were analyzed. Patients with aHUS with the absence of specific treatment, those with aHUS with eculizumab prophylactic treatment, and those with aHUS with eculizumab treatment after transplantation.

**No eculizumab use.** Patients who did not receive any dose of specific complement-inhibitor.

**Prophylactic eculizumab.** Patients with a pre-transplant aHUS diagnosis who underwent kidney transplantation during eculizumab treatment or who started the medication before graft reperfusion.

**Eculizumab treatment.** Patients who presented TMA in the post-transplant period and received eculizumab for the treatment of recurrence.

## Outcome

The primary outcome analyzed was the recurrence of TMA leading to graft loss. Secondary outcomes analyzed were the number of rejection episodes and death.

## Statistical analysis

Continuous variables were presented as medians and percentiles (25 and 75%) and categorical variables were expressed as numbers and percentages. Comparisons between groups for continuous variables were made using the Kruskal-Wallis test and for categorical ones, we used the chi-square test. Adjustment test was used for multiple comparisons: False discovery rate correction for multiple testing according to Benjamini & Hochberg (1995) [20].

Kaplan-Meier analysis was constructed to estimate the effect of treatment on the incidence of the outcome (TMA graft loss). Hazard ratio estimates were performed using Cox regression. The validity of the Cox model was assessed by the Schoenfeld residue analysis. The analyzes were made with the software R version 4.0.2 and the survival and gtsummary packages.

## Results

Of the 118 kidney transplant centers in Brazil, 6 centers agreed to participate and completed the regulatory procedures and provided data from 45 patients with a presumed diagnosis of aHUS. Of the 45 patients, 7 cases were excluded because they did not fulfill the aHUS criteria (Fig 1), that is, presence of antibody-mediated rejection (n = 3), thrombotic thrombocytopenic purpura (n = 1), and absence of TMA findings at baseline kidney biopsy (n = 3). Thus, our final sample included 38 patients (Fig 1).

The median age was 30 (24–40) years, with a predominance of females (63%), the most frequent kidney disease was undetermined, and the majority of aHUS patients were transplanted with a deceased donor (68%). The majority were induced with thymoglobulin (61%) and the maintenance immunosuppression was the combination of tacrolimus associated with mycophenolate and prednisone (75%). The median year of aHUS presentation was the year 2015 (2013, 2017) (Table 1).

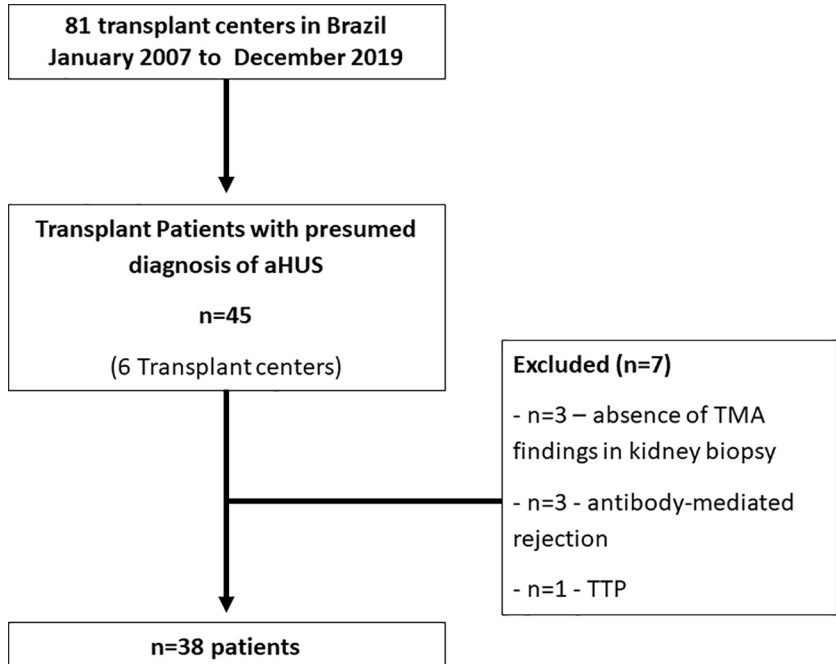

aHUS: Atypical Hemolytic Uremic Syndrome;     TTP: thrombotic thrombocytopenic purpura
TMA: Thrombotic microangiopathy

**Fig 1. Flowchart study.**

**Table 1. Demographic data of Brazilian aHUS cohort patients in renal transplantation.**

| Features | N = 38 |
|---|---|
| **Age (years)** | 30(24.40) |
| **Sex (n,%)** | |
| Female | 24 (63%) |
| Male | 14 (37%) |
| **Ethnicity (n,%)** | |
| White | 22(61%) |
| Black/Pardo | 14(39%) |
| Missing | 2 |
| **Year at aHUS diagnosis** | 2,015(2,013, 2,017) |
| **Panel Reactive Antibody (%)** | 0 (0.20) |
| Missing | 4 |
| **Underlying kidney disease (n,%)** | |
| Glomerulonephritis | 5 (14%) |
| Indeterminate | 17 (46%) |
| C3 Nephropathy | 2 (5.4%) |
| aHUS | 11 (30%) |
| Other | 2 (5.4%) |
| Missing | 1 |
| **Dialysis method (n, %)** | |
| Preemptive | 1 (2.6%) |
| Peritoneal | 5 (13%) |
| Hemodialysis | 32 (84%) |
| **Time on Dialysis (months)** | 22 (11.35) |
| Missing | 1 |
| **Transplant Donor (n,%)** | |
| Deceased | 26 (68%) |
| Living | 12(32%) |
| **Donor Age (years)** | 42 (28.50) |
| Missing | 3 |
| **Cause of death donor (n, %)** | |
| Cerebrovascular/Stroke | 12 (48%) |
| Head Trauma | 10 (40%) |
| Other | 3 (12%) |
| Missing | 13 |
| **Induction Therapy (n, %)** | |
| Thymoglobulin | 22 (61%) |
| Basiliximab | 8 (22%) |
| Without induction | 6 (17%) |
| Missing | 2 |
| **Eculizumab Induction (n, %)** | 7 (19%) |
| Missing | 1 |
| **Immunosuppression (n, %)** | |
| Tac + MFS + P | 27 (75%) |
| Tac + imTOR + P | 7 (19%) |
| Outher | 2 (5.6%) |
| Missing | 2 |
| **TREATMENT** | |

(*Continued*)

Table 1. (Continued)

| Features | N = 38 |
|---|---|
| **Plasmapheresis (n, %)** | 16 (42%) |
| No plasmapheresis and no eculizumab | 5 (13%) |
| Only Eculizumab | 17 (45%) |
| Only plasmapheresis | 6 (15%) |
| Plasmapheresis and eculizumab | 10 (26%) |
| **Eculizumab (n, %)** | |
| No | 11 (19%) |
| Prophylactic | 10 (26%) |
| Treatment | 17 (45%) |
| **Initial accesss to Eculizumab** | |
| No access to medication | 11 (29%) |
| Compassionate access program | 24 (63%) |
| Brazilian judicial system | 3 (7,9%) |
| OUTCOME | |
| **Acute Rejection (n, %)** | 12 (32%) |
| **TMA graft loss (n, %)** | 12 (32%) |
| **One-year Death** | 3 (8%) |
| **All-Time Death (n, %)** | 6 (17%) |

Continuous variables expressed as the median and interquartile range (25 and 75%)

aHUS: atypical Hemolytic Uremic Syndrome; Tac: Tacrolimus; MFS: Mycophenolate sodium; P: Prednisone; imTOR: mTOR inhibitors; TMA: thrombotic microangiopathy.

Of the total, 42% underwent plasmapheresis treatment and 71% received eculizumab (26% received plasmapheresis and eculizumab). According to eculizumab stratification, 11 patients were in the No Eculizumab use, 10 were in the Prophylactic Group and 17 were in the Therapeutic Group. The TMA graft loss occurred in 32% (n = 12/38) of the cases, and 17% (n = 6/38) of patients died during the study period (**Table 1**).

The analysis of genetic variants was performed in 19 patients (50% of the sample). Negative results were found in 26% of the patients. Among the positive findings, the most frequent variant was detected in Factor H (26%). The variants related to factor H proteins (CFHR5 and CFHR1-CFHR3), Factor I and Thrombomodulin had a frequency of 10.5% each (**Table 2, S1 Fig**). Detailed analyses of the variants are described in the (**S1 Table**). The frequency of variants divided by groups was provided in the (**S2 Table**).

## TMA recurrence

When analyzing the groups, 91% (n = 10/11) of patients in the group that did not receive eculizumab lost the graft due to the aHUS recurrence. In contrast, both groups that received the eculizumab had lower rates of graft loss, 10% (n = 1/10) in the Prophylactic Group and 5.9% (n = 1/17) in the Therapeutic Group (p <0.001) (**Table 3**). In the prophylactic group one patient evolved to graft loss due to TMA after hospitalization due to arteriovenous fistula thrombosis and had a delay of eculizumab infusion at that time. In the treatment group one patient lost the graft, probably due to a delay in starting eculizumab treatment, which was performed more than 120 days after diagnosis.

The acute rejection occurred in 67% of the patients who did not use the eculizumab compared with 0 and 35% in the Prophylactic and Therapeutic groups, respectively (p = 0.021)

**Table 2. Genetic variants of Brazilian aHUS cohort patients in renal transplantation.**

| | | n = 38 | N = 19 |
|---|---|---|---|
| | | **Total cases** | **Genetic analysis performed** |
| | | (% total) | (% cases performed) |
| **Genetic Test Not Performed** | | 19 (50%) | |
| **Genetic Test** | No variants found | 5 (13%) | 5 (26.3%) |
| | CFH[&] | 5 (13%) | 5 (26.3%) |
| | CFHR5 | 2 (5.3%) | 2 (10.5%) |
| | CFHR1-CFHR3 | 2 (5.3%) | 2 (10.5%) |
| | CFI[*] | 2 (5.3%) | 2 (10.5%) |
| | TBHD[+] | 2 (5.3%) | 2 (10.5%) |
| | C3 | 1 (2.6%) | 1 (5.2%) |

& one case of the CFH variant was associated with CFI

* in one case of the CFI variant it was associated with the CFB variant and in another case the CFI was associated with CFHR1-CFHR5; + one case of the TBHD variant was associated with CFHR5 and another case associated with PLG.

(**Table 3**). The use of eculizumab occurred more recently (2015 for the therapeutic group and 2017 for the prophylactic group) compared with the group that did not receive the drug (2013, p = 0.07) (**Table 3**).

The incidence of TMA graft loss at 1000 days was respectively in the therapeutic, prophylaxis and no eculizumab groups: 7.6%, 10%, and 86.4%, p <0.0001 (**Fig 2**). The hazard ratio graft loss was 0.07 [0.01–0.55], p = 0.012 in the eculizumab Prophylactic group and 0.04 [0.00–0.28], p = 0.002 in Therapeutic group, much lower compared with the eculizumab non-use group.

The cumulative survival at 1000 days was 100% in the Prophylaxis, 70% in Therapeutic group, and 40% in the no eculizumab group (p = 0.13) (**Fig 3**). The cause of death in the eculizumab treatment were: hemorrhagic shock (n = 1) and septic shock (n = 3). The cause of death in no eculizumab treatment was TMA recurrence (n = 2) (**S3 Table**).

### Time of aHUS diagnosis

Regarding the year of aHUS diagnosis, two patients in the no eculizumab group were diagnosed before the year of 2010. Patients in the eculizumab groups (prophylactic or therapeutic) were diagnosed after the year 2010 and the majority of the cases evolved without TMA recurrence (**Fig 4**).

### Discussion

The present study evaluated TMA recurrence in a large cohort including aHUS patients after kidney transplantation in Brazil. The Brazilian aHUS cohort was composed predominantly of women (63%) and young adults (30 years), which is a similar finding reported by other studies [6, 21, 22]. Before the advent of complement inhibitors, the treatment of these patients was limited to plasmapheresis and/or plasma infusion that had shown unsatisfactory results [23]. Confirming these observations, in another Brazilian cohort of TMA patients after transplantation, the graft survival was greatly reduced without the use of specific complement-inhibitors [24]. However, the inclusion of eculizumab as a therapeutic or prophylactic option after transplantation has been significantly improving graft survival [7, 8] by preventing the recurrence of TMA [17, 22]. We demonstrated that the use of the eculizumab in prophylaxis or treatment was associated with a significant reduction in TMA graft loss.

**Table 3. Comparison of demographic and outcome in the Brazilian aHUS cohort patients according to the treatment (without the use of Eculizumab, prophylactic Eculizumab, and Eculizumab treatment).**

| Features | Eculizumab No (N = 11) | Eculizumab Prophylactic (N = 10) | Eculizumab Treatment (N = 17) | P adjusted * |
|---|---|---|---|---|
| **Age (years)** | 29(28,45) | 26(19,32) | 31(26,38) | 0.3 |
| **Sex (n,%)** | | | | 0.6 |
| Female | 6(55%) | 8(80%) | 10(59%) | |
| Male | 5(54%) | 2(20%) | 7(41%) | |
| **Ethnicity (n,%)** | | | | 0.3 |
| White | 6(55%) | 8(89%) | 8(50%) | |
| Black/Pardo | 5(54%) | 1(11%) | 8(50%) | |
| Missing | 0 | 1 | 1 | |
| **Panel Reactive Antibody (%)** | 0(0.4) | 0(0.0) | 0(0.22) | 0.7 |
| Missing | 2 | 1 | 1 | |
| **Year at aHUS diagnosis** | 2,013 (2,011, 2,014) | 2,017 (2,015, 2,017) | 2,015 (2,014, 2,016) | 0.072 |
| **Underlying kidney disease (n,%)** | | | | 0.007 |
| Glomerulonephritis | 2(18%) | 0(0%) | 3(19%) | |
| Indeterminate | 6(55%) | 1(10%) | 10(62%) | |
| C3 Nephropathy | 2(18%) | 0(0%) | 0(0%) | |
| Others | 0(0%) | 1(10%) | 1(6.2%) | |
| aHUS | 1(9.1%) | 8(80%) | 2(12%) | |
| Missing | 0 | 0 | 1 | |
| **Dialysis method (n,%)** | | | | 0.8 |
| Preemptive | 1(9.1%) | 0(0%) | 0(0%) | |
| Peritoneal | 1(9.1%) | 2(20%) | 2(12%) | |
| Hemodialysis | 9(82%) | 8(80%) | 15(88%) | |
| **Time on Dialysis (months) (n,%)** | 11(9.21) | 26(18.56) | 22(19.35) | 0.2 |
| Missing | 1 | 0 | 0 | |
| **Transplant Donor (n,%)** | | | | 0.13 |
| Deceased | 5(45%) | 7(70%) | 14(82%) | |
| Living | 6(55%) | 3(30%) | 3(18%) | |
| **Donor Age (years)** | 50(43.54) | 34(18.45) | 39(33.47) | 0.3 |
| Missing | 3 | 0 | 0 | |
| **Cause of death donor (n,%)** | | | | 0.3 |
| Cerebrovascular/Stroke | 3(75%) | 1(14%) | 8(57%) | |
| Head Trauma | 1(25%) | 5(71%) | 4(29%) | |
| Others | 0(0%) | 1(14%) | 2(14%) | |
| Missing | 7 | 3 | 3 | |
| **Induction Therapy (n,%)** | | | | 0.10 |
| Thymoglobulin | 2(22%) | 8(80%) | 12(71%) | |
| Basiliximab | 3(33%) | 2(20%) | 3(18%) | |
| Without induction | 4(44%) | 0(0%) | 2(12%) | |
| Missing | 2 | 0 | 0 | |
| **Immunosuppression (n,%)** | | | | 0.021 |
| Tac + MFS + P | 9(100%) | 10(100%) | 8(47%) | |
| Tac + imTOR + P | 0(0%) | 0(0%) | 7(41%) | |
| Othres | 0(0%) | 0(0%) | 2(12%) | |
| Missing | 2 | 0 | 0 | |

*(Continued)*

**Table 3.** (Continued)

| Features | Eculizumab No (N = 11) | Eculizumab Prophylactic (N = 10) | Eculizumab Treatment (N = 17) | P adjusted * |
|---|---|---|---|---|
| **Plasmapheresis (n,%)** | 5(56%) | 3(30%) | 5(38%) | 0.7 |
| Missing | 2 | 0 | 4 | |
| **Acute Rejection** | 6(67%) | 0(0%) | 6(35%) | 0.021 |
| **TMA graft loss (n,%)** | 10(91%) | 1(10%) | 1(5.9%) | <0.001 |
| **One-Year Death (n,%)** | 1(9%) | 0(0%) | 2(12%) | 0.55 |
| **All-Time Death (n,%)** | 2(20%) | 0(0%) | 4(27%) | 0.4 |

Continuous variables expressed as the median and interquartile range (25 and 75%)

aHUS: atypical Hemolytic Uremic Syndrome; Tac: Tacrolimus; MFS: Mycophenolate sodium; P: Prednisone; imTOR: mTOR inhibitors; TMA: thrombotic microangiopathy.

* p adjusted for multiple comparisons.

In our cohort, 42% of the patients underwent plasmapheresis, but the use of the eculizumab was the best predictor associated with a lower risk of TMA recurrence. A French cohort conducted by Zuber *et al.* had demonstrated that eculizumab had revolutionized clinical outcomes after kidney transplantation, practically abolishing graft losses due to the TMA recurrence [25]. Similarly, Siedlecki et al. [26] from the Global aHUS Registry, showed that a delay in eculizumab treatment is associated with an increased risk of dialysis after transplantation and reduced allograft function. Thus, according to KDIGO [1], patients diagnosed with aHUS should be treated primarily with eculizumab aiming to prevent TMA recurrence. Despite these recommendations, the use of eculizumab in middle-income countries is still a challenge.

In Brazil, the first use of prophylactic eculizumab reported in our cohort occurred in 2011 through compassionate use [27]. Almost all cases in this cohort had initial access to medication by compassionate access program (92% of cases) followed by the access to judicialization. The eculizumab was registered by the Brazilian Health Regulatory Agency (ANVISA) in March 2017 [28]. In 2018, a specialized evaluation was requested from the Brazilian Institute of Health (CONITEC) to incorporate the medication into the public health system. However, in 2019 CONITEC concluded that eculizumab was not suitable to be incorporated into the public health system due to the high cost of this therapy [29]. For this reason, currently, in Brazil, the only way to access eculizumab is through judicialization, which in this country is a lengthy process and that delays the beginning of the treatment in months [30]. Despite these difficulties, the higher rates of eculizumab use in the present cohort (71%) possibly reflect on the structure of the transplant services in Brazil that is linked to university centers, which could accelerate the investigation of rare diseases. Another contributing factor to eculizumab availability was the access by a compassionate access program that was available until the year 2019.

The impact of this information is relevant to Brazil, which is ranked at the first position in the number of transplants performed by the Brazilian public health system (SUS) [31] and reserves for this purpose an annual budget around R$1 billion [31]. In this context, the use of a high-cost medication such as eculizumab, which acquisition expenditure by the Brazilian government in the year 2016 was R$624,621,376.84 (60% of the annual budget destined for transplantation) may jeopardize the sustainability of the health program [29]. Thus, an ethical dilemma is imposed on the use of a high-cost medication, but that shows great efficacy. Cost-effectiveness studies have shown that the long-term use of eculizumab was not cost-effective in transplantation. In contrast, short-term use of eculizumab upon recurrence has proven to be

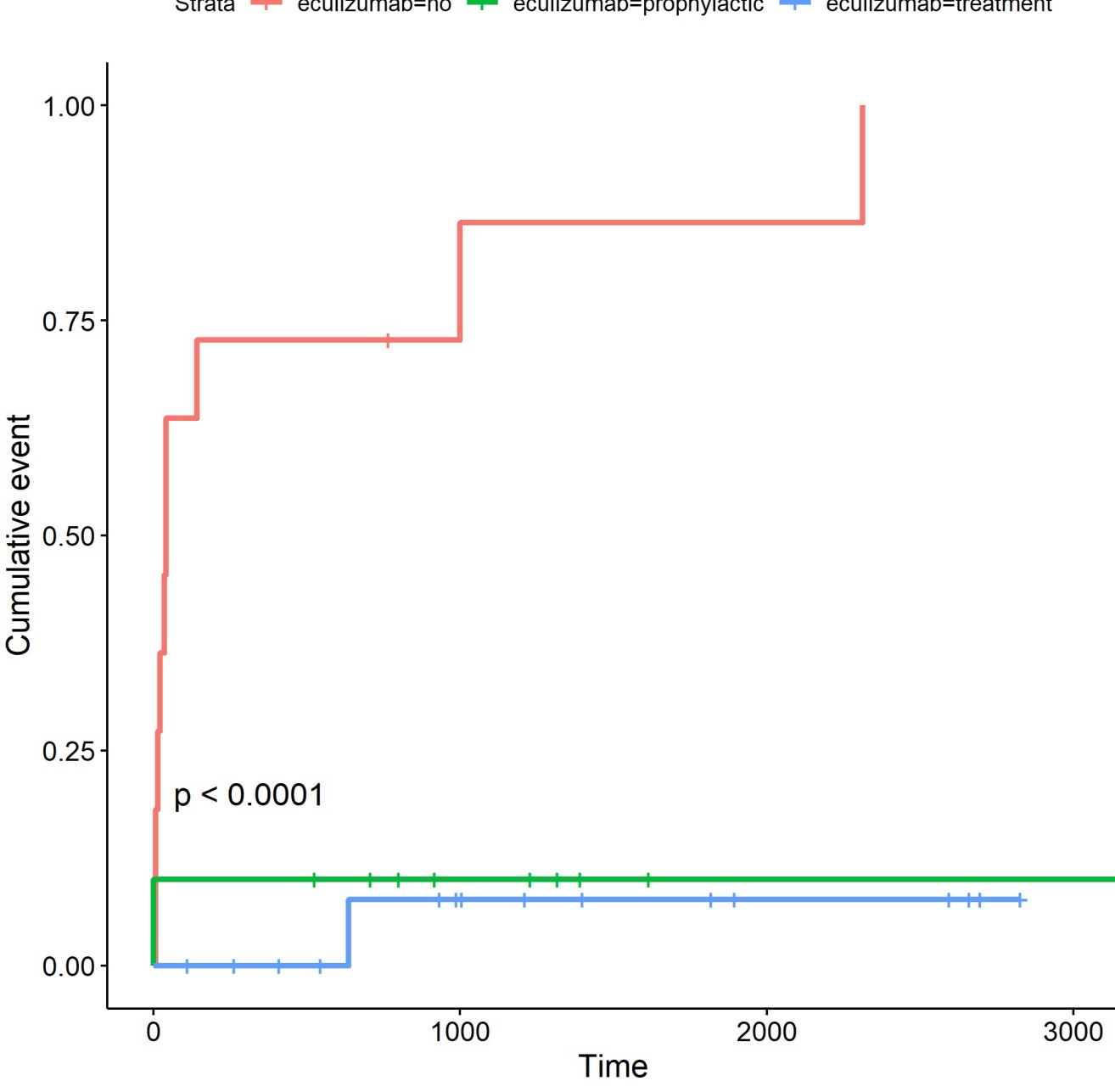

**Fig 2. Cumulative incidence of TMA graft loss (days) in the Brazilian aHUS cohort patients divided by groups: Not received eculizumab (red), prophylactic eculizumab (green), and eculizumab treatment (blue).**

cost-effective and has resulted in an incremental quality-adjusted life year (QALY) of 9.55 [32]. Despite the uncertainty regarding the safety of eculizumab interruption after transplantation [33] it seems rational to propose its use for a short-time period compared to providing no access to the treatment with eculizumab. This strategy may benefit the access of eculizumab without resulting in an excessive cost to the public health system, especially in low-income countries. However, to date, the best strategy to definided eculizumab interruption was based on genetic analysis [25, 34]. The cost-effectiveness of accessing eculizumab for kidney transplant patients should be based on genetic analysis, thus, it is necessary to include the genetic

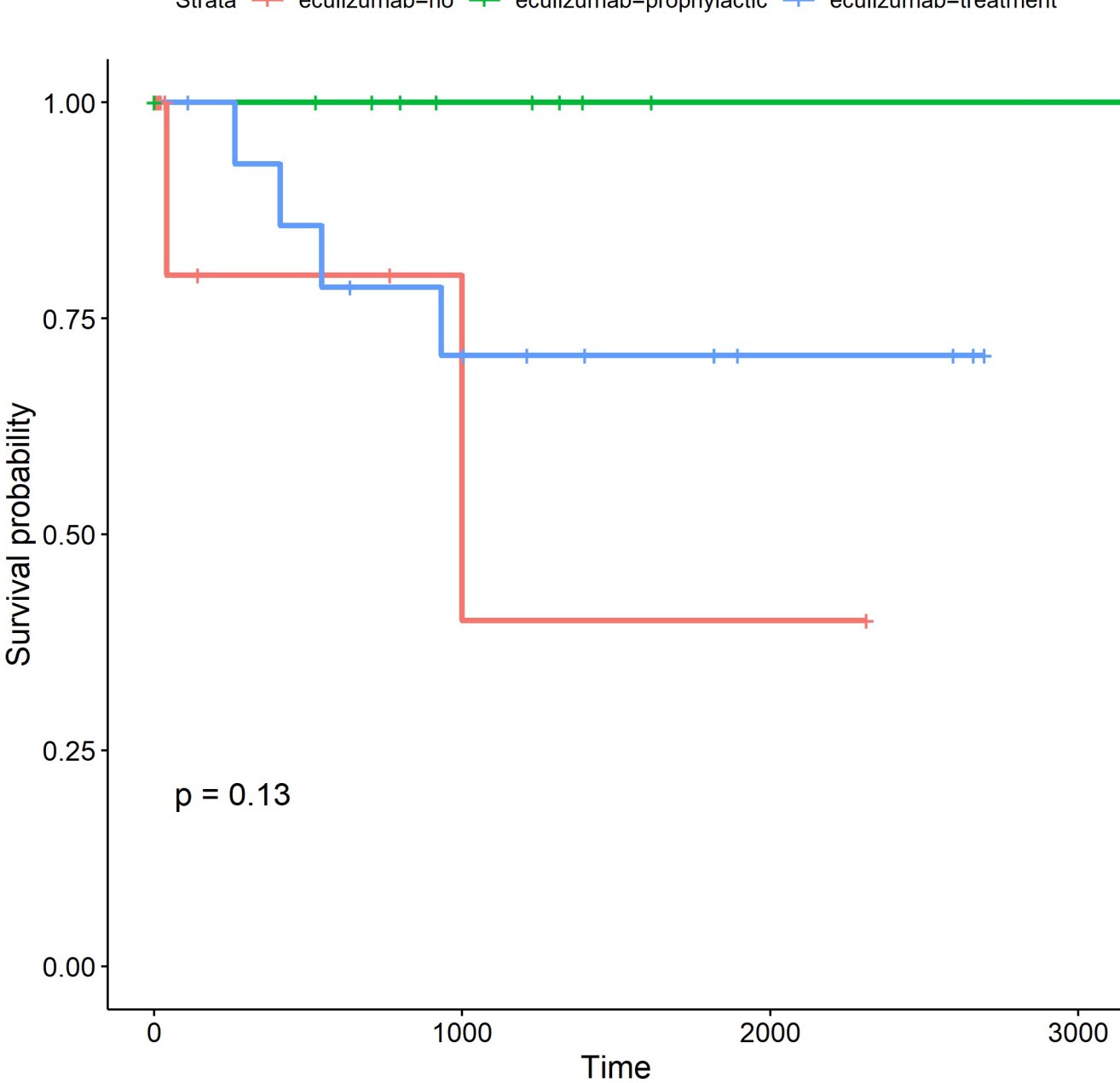

**Fig 3. Cumulative survival (days) in the Brazilian aHUS cohort patients divided by groups: Not received eculizumab (red), prophylactic eculizumab (green), and eculizumab treatment (blue).**

profile for patients with suspicion for aHUS. In those classified at high to moderate risk according to KDIGO 2017 [1] we strongly consider prophylaxis. As the opposite, we consider discontinuing the use in those without identified genetic variants, as well as the saving of 32 million euros realized by the STOPECU study [34] after the discontinuation of 55 patients with an average of 24 months of follow-up.

The mortality observed in the no eculizumab group was higher and earlier after transplantation, this is possibly related to the uncontrolled activity of the complement system. Intermediate mortality was observed in the therapeutic group that may be justified by the delay

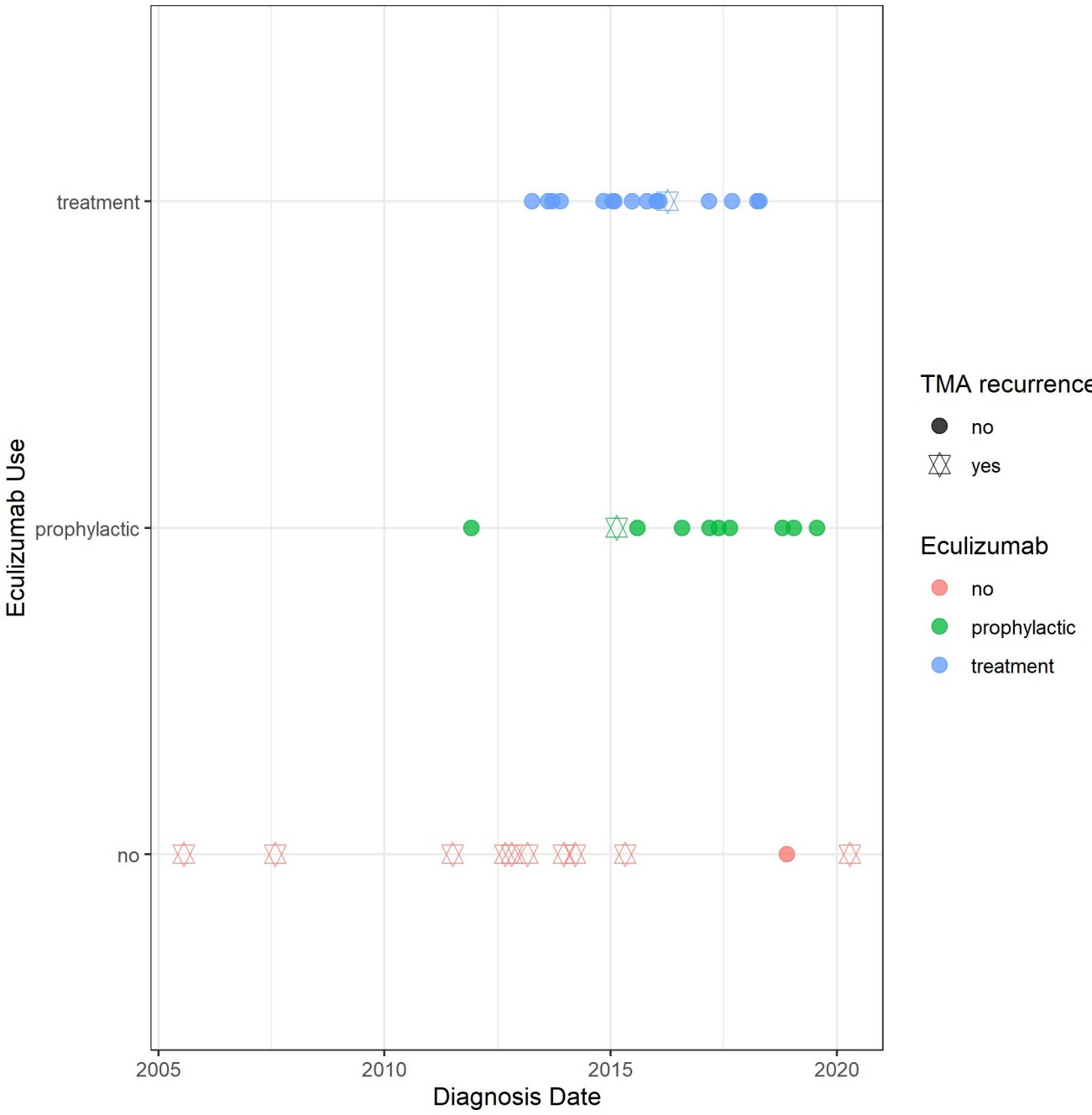

**Fig 4. Outcomes of the Brazilian aHUS cohort patients divided by year of the aHUS diagnosis.** Colors: not received eculizumab (red), prophylactic eculizumab (green) and eculizumab treatment (blue).

between diagnosis and medication availability, a similar finding to what was reported in the aHUS registry [26]. The best survival (100%) was observed in the prophylaxis group that received the medication in a planned way. There were no infections secondary to encapsulated organisms as a result of eculizumab treatment.

Another point of this study was the high rate of acute rejection in the first year of transplantation, reaching up to 67% in the group that did not receive eculizumab and 35% in the

eculizumab treatment group. This fact can be attributed to the conversion or reduction of immunosuppression at the time of the clinical presentation of TMA, such as the suspension of the calcineurin inhibitor [17]. Corroborating with this fact, in the prophylaxis group, where changes in immunosuppression were not performed, there were no episodes of rejection. Regarding the severity of the rejection episodes, although there is no biopsy-proven of all cases, the response to treatment with corticosteroids, and the absence of anti-donor antibodies reinforce the fact that they were probably T-cell-mediated rejections.

Another important piece of information was the genetic analysis that can be used to determine the genetic profile of the Brazilian population recognized by high rates of miscegenation [35]. The difficulty to prescribe genetic tests in a middle-income country as Brazil is reflected in this series in which 50% of the sample did not have access to genetics. Of the cases that underwent genetic analysis tests, in 26% no variant was found, similar to the literature that reports rates of 30–40% [36]. Of those cases that had variants, the most frequent was in Factor H, a similar result reported by other authors [11, 37]. We found a higher frequency of variants related to the complement system such as CFHR1-CFHR3 and CFHR5 in this cohort, similarly to other Brazilian cohorts such as Palma et al. [21], Andrade et al. e Ernandes-Neto [33]. Additionally, two cases were found with variants related to the coagulation system (thrombomodulin) related to the aHUS [38].

## Study limitations

This is an observational study that has design limitations of a retrospective analysis. The optimal design for comparing different treatments would be a randomized study. However, we were unable to adopt such a design given the rare characteristic of aHUS. The access to complement inhibitor medication was not homogeneous in each center, varying according to the local protocol and availability of the compassionate access program. The group that did not receive the eculizumab was different from the other two groups. These patients received a kidney transplant in older time and had higher rates of acute rejection and lower use of induction therapy. These may compromise the results of the graft survival analysis. Additionally, we could not confirm the episodes of acute rejection with a biopsy proven.

As we evaluated a retrospective cohort, cases from different timelines were included. Despite the large period considered in this study, the majority of the cases (95%) was contemporarily occurring after the year of 2011. There were only two patients (5% of the sample) diagnosed before 2010, where there was no possibility of treatment with complement inhibitors. In genetic analysis, despite the presence of variants related to Factor H (CFHR1-CFHR3 and CFHR5), there was no possibility of dosing the anti-factor H antibody because these tests are not available in Brazil. Due to the sample size, it was not possible to add other confounding variables in the Cox analysis or to perform stratifications by the type of genetic variant.

Despite these limitations, to our knowledge, this is the first study of the Brazilian aHUS cohort in kidney transplantation. In a country with limited resources such as Brazil and with a huge public transplant program, analyzing the eculizumab treatment is fundamental to define public health policies.

## Conclusion

The TMA graft loss in the absence of a specific complement-inhibitor was high in the Brazilian cohort of kidney transplant patients. This reinforces the need for eculizumab use in kidney transplant patients diagnosed with aHUS as a prophylactic or therapeutic approach. There was a tendency of better results due to the use of eculizumab prophylactic, which resulted in better patient survival rate. Further cost-effectiveness and discontinuity studies are warranted to

assess the financial impact on the eculizumab use in kidney transplantation, especially in low-medium income countries.

## Supporting information

**S1 Fig. Results of genetic analysis in cases submitted to genetic analysis (n = 19) in the Brazilian aHUS cohort in kidney transplantation.**
(TIF)

**S1 Table. The demographic data and complete genetic analysis in the Brazilian aHUS cohort in kidney transplantation.**
(DOCX)

**S2 Table. The frequency of variants divided by groups: No eculizumab use, eculizumab treatment, and prophylactic eculizumab in the Brazilian aHUS cohort in kidney transplantation.**
(DOCX)

**S3 Table. Cause of mortality in the Brazilian aHUS cohort in kidney transplantation.**
(DOCX)

## Author Contributions

**Conceptualization:** Lilian Monteiro Pereira Palma, Miguel Ernandes Neto, Silvana Maria Carvalho Miranda, Luis Gustavo Modelli de Andrade.

**Formal analysis:** Luis Gustavo Modelli de Andrade.

**Investigation:** Hong Si Nga, Miguel Ernandes Neto, Ida Maria Maximina Fernandes-Charpiot, Valter Duro Garcia, Roger Kist, Silvana Maria Carvalho Miranda, Pedro Augusto Macedo de Souza, Gerson Marques Pereira, Jr.

**Methodology:** Hong Si Nga.

**Project administration:** Hong Si Nga.

**Supervision:** Luis Gustavo Modelli de Andrade.

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
