## [Decision Letter · Decision Letter 0]

24 Aug 2021

PONE-D-21-24463

Thrombotic microangiopathy after kidney transplantation: analysis of the Brazilian Atypical Hemolytic Uremic Syndrome cohort

PLOS ONE

Dear Dr. de Andrade,

Thank you for submitting your manuscript to PLOS ONE. After careful consideration, we feel that it has merit but does not fully meet PLOS ONE’s publication criteria as it currently stands. Therefore, we invite you to submit a revised version of the manuscript that addresses the points raised during the review process.

The manuscript focuses on a topic of potential interest. However, the study has several drawbacks that should be addressed before reaching sound conclusion. To mention some of them, i) concern about the fact that there are 118 kidney transplant centers across Brazil, but only 6 accepted to include their patients in this retrospective study; ii) unclear how many patients were  kidney transplanted across Brazil and how many in the 6 centers enrolled during the period of 12 years of this retrospective study; iii) need to provide donor-specific alloantibodies (DSAs) at the time of TMA recurrence; iv) need to provide the cause of allograft loss across the 3 groups; v) concern about the fact that the results are confirmatory in nature; vi) unclear how many patients lost their graft in the Prophylactic and Treatment groups; vii) need to provide detail of the genetic analysis for the group treated with eculizumab versus the group that was untreated; viii) concern about the fact that the cause of end-stage renal disease is undetermined in almost half of patients; ix) concern about the fact that the complement genetic data are available only in half of included patients and the distribution of complement gene variants within the three groups of patients is unknown; x) concern about the high incidence of acute rejection in the group of patients who did not receive eculizumab, which is a confounding factor for the analysis of graft loss; xi) need to provide the causes of death in patients who did not receive prophylactic eculizumab; xii) need to focus the discussion more on the results; xiii) need to clarify that only prophylactic eculizumab prevents recurrence; xiv) need to underline that the differences in the survival probabilities between groups are not statistically significant probably due to the small sample size.

We look forward to receiving your revised manuscript.

Kind regards,

Giuseppe Remuzzi

Academic Editor

PLOS ONE

Journal Requirements:

2. Please provide additional details regarding participant consent. In the Methods section, please ensure that you have specified (1) whether consent was informed and (2) what type you obtained (for instance, written or verbal). If your study included minors, state whether you obtained consent from parents or guardians. If the need for consent was waived by the ethics committee, please include this information.

"L.M.P.P., S.M.C.M, and L.G.M.A. have receive fees as lectures for Alexion Pharmaceuticals, Brazil. All the other authors declared no competing interests. "

We note that you received funding from a commercial source: "Alexion Pharmaceuticals".

Reviewers' comments:

Reviewer's Responses to Questions

**Comments to the Author**

1. Is the manuscript technically sound, and do the data support the conclusions?

Reviewer #1: Partly

Reviewer #2: Yes

Reviewer #3: Yes

Reviewer #4: Yes

2. Has the statistical analysis been performed appropriately and rigorously? 

Reviewer #1: No

Reviewer #2: Yes

Reviewer #3: Yes

Reviewer #4: Yes

3. Have the authors made all data underlying the findings in their manuscript fully available?

Reviewer #1: Yes

Reviewer #2: Yes

Reviewer #3: Yes

Reviewer #4: Yes

4. Is the manuscript presented in an intelligible fashion and written in standard English?

Reviewer #1: Yes

Reviewer #2: No

Reviewer #3: Yes

Reviewer #4: Yes

5. Review Comments to the Author

Reviewer #1: The authors report on their experience with renal transplantation in patients with a presumed or confirmed diagnosis of aHUS in 6 Brazilian transplant centres. They included 38 patients, a significant number of patients in regard of the rarity of the disease. For the purpose of the analysis, the patients were divided into three groups: no treatment with eculizumab, curative or prophylactic treatment with eculizumab.

The authors conclude that eculizumab use (particularly as prophylaxis) significantly improved graft but also patients’ survival. The novelty of the data is rather limited, and previous larger studies have shown similar trends for graft loss due to aHUS recurrence.

I have the following major comments:

a) The cause of end-stage renal disease is undetermined in a large proportion (almost half) of patients.

b) The complement genetic data are available in only 19/38 included patients and the distribution of complement gene variants within the three groups of patients is unknown. This is a crucial point as complement variants are a major predictor for aHUS relapse after renal transplantation. The detected variants should also be classified as pathogenic, likely pathogenic or variants of unknown significance.

c) The high incidence of acute rejection (67%) in the group of patients who did not receive eculizumab is clearly a confounding factor for the analysis of graft loss. Besides, this group included the highest rate of living donors (55%) and no induction therapy in 44% of cases. It is in sharp contrast with the Dutch experience with living donation without prophylactic eculizumab in aHUS pateunts.

d) What were the causes of death in patients who did not receive prophylactic eculizumab? A rate of 20-27% of death in two groups of pateints is unusually high and hardly explained by a potential aHUS recurrence.

d) The discussion is more focused on the accessibility to eculizumab in Brazil rather than on the results of the study. A cost-effective use of eculizumab in renal transplantation is based in good part on genetic analysis, and this point needs to be more developed (see reference Zuber et al. JASN).

Reviewer #2: This was a multicenter retrospective cohort study on 38 patients with ESKD due to aHUS who underwent kidney transplant. Patients were stratified into three groups: no use of eculizumab (No Eculizumab Group), use of eculizumab for treatment of after transplantation TMA (Therapeutic Group), and use of eculizumab for prophylaxis of aHUS recurrence (Prophylatic Group). In the No Eculizumab Group (n=11), there was a 91% graft loss due to the TMA. The hazard ratio of TMA graft loss was 0.07 [0.01-0.55], p = 0.012 in the eculizumab Prophylatic Group and 0.04 [0.00 - 0.28], p =0.002 in the eculizumab Therapeutic Group. The authors conclude that TMA graft loss in the absence of a specific complement-inhibitor was higher among the Brazilian cohort of kidney transplant patients. This finding reinforces the need of eculizumab use for treatment of aHUS kidney transplant patients.

This is a nice retrospective study in a group of patients with an ultrarare disease.

Comments:

1. How many patients lost their graph in the Prophylactic and Treatment group? The manuscript just gives %

2. Please clarify: “42% underwent plasmapheresis and 79% received eculizumab”. This is not clear for total number of patients that received eculizumab was 27/38 = 71%

3. “TMA graft loss occurred in 32% of cases and 17% died” please give actual numbers.

4. Please provide detail of the genetic analysis for the group treated with Eculizumab versus the group that was untreated.

5. Table 2. 18 + 19 = 37. There is 1 patient missing.

6. Please provide details of the Eculizumab regime used. Was Eculizumab ever discontinued post-transplant? What was the criteria used? What was the outcome in patients that discontinued Eculizumab?

7. There are a few typos/grammar mistakes that need to be fixed.

Reviewer #3: This interesting article highlights the efficacy of complement blockage in preventing and treating the aHUS recurrence after kidney transplantation. The results are consistent with those previously published by Zuber and Siedlecky, as you mentioned in the paper. It would had been very interesting if the genetic study of all patients was available. In spite of that, is important to remark that the patients with CFI mutations had TMA recurrence only if they have combined mutations

- P26 line 8:

"However, the inclusion of the eculizumab as a therapeutic option after transplantation has been significantly improving graft surviva by preventing the recurrence of TMA"

In this line, only prophylactic eculizumab prevents recurrence. Please, clarify.

- In the discussion section, you have mentioned the probable causes associated to the high rate of acute rejection.However, you haven't showed data related to this assumption in the results section. Could you give more information?

- The differences in the survival probabilities between groups are not statistically significant, probably due to the small sample size. Please, remark this finding

Reviewer #4: I reviewed the manuscript PONE-D-21-24463 titled "Thrombotic microangiopathy after kidney transplantation: analysis of the Brazilian Atypical Hemolytic Uremic Syndrome cohort". This is a retrospective multicentris small cohort study that included only 38 aHUS KTx.

Because the results are confirmatory in nature the papzr deserves not more than a Letter to the Editor.

There are 118 kidney transplant centers across Brazil but only ..... 6 accepted to include their patients in this retrospective study that took place between January 2007 and December 2019. during that period of 12 years how many patients were kidney transplanted across Brazil and how many in these 6 centers?

Were donor-specific alloantibodies (DSAs) looked for at the time of TMA recurrence, i.e., in many patients the cause of ESRD was not aHUS?

Across the 3 groups what were the causes of allograft loss?

6. PLOS authors have the option to publish the peer review history of their article (what does this mean?). If published, this will include your full peer review and any attached files.

Reviewer #1: No

Reviewer #2: **Yes: **Fernando C Fervenza

Reviewer #3: No

Reviewer #4: No

---

## [Author Response · Author response to Decision Letter 0]

3 Sep 2021

Reviewer #1: The authors report on their experience with renal transplantation in patients with a presumed or confirmed diagnosis of aHUS in 6 Brazilian transplant centres. They included 38 patients, a significant number of patients in regard of the rarity of the disease. For the purpose of the analysis, the patients were divided into three groups: no treatment with eculizumab, curative or prophylactic treatment with eculizumab.

The authors conclude that eculizumab use (particularly as prophylaxis) significantly improved graft but also patients’ survival. The novelty of the data is rather limited, and previous larger studies have shown similar trends for graft loss due to aHUS recurrence.

I have the following major comments:

a) The cause of end-stage renal disease is undetermined in a large proportion (almost half) of patients.

In the initial pre-transplant evaluation, the diagnosis of kidney disease was undetermined. However, at this time, the aHUS evaluation was not carried out because they had no diagnostic suspicion. These patients evolved with aHUS after transplantation and after that were treated with eculizumab (eculizumab treatment group). In fact, these groups of patients were probably aHUS undiagnosed prior kidney transplant. 

b) The complement genetic data are available in only 19/38 included patients and the distribution of complement gene variants within the three groups of patients is unknown. This is a crucial point as complement variants are a major predictor for aHUS relapse after renal transplantation. The detected variants should also be classified as pathogenic, likely pathogenic or variants of unknown significance.

Thank you for the very important comment. We adjusted the table with this requested information – (supplementary S2 Table and S3 Table).

c) The high incidence of acute rejection (67%) in the group of patients who did not receive eculizumab is clearly a confounding factor for the analysis of graft loss. Besides, this group included the highest rate of living donors (55%) and no induction therapy in 44% of cases. It is in sharp contrast with the Dutch experience with living donation without prophylactic eculizumab in aHUS patients.

We agree with the reviewer. The group that did not receive the eculizumab was different from the other two groups. These patients received a kidney transplant in older time and had higher rates of acute rejection and lower use of induction therapy. These may compromise the results of the graft survival analysis. We included these in the limitations. However, we believe that this limitation did not compromise the primary study aim that was the TMA recurrence. 

d) What were the causes of death in patients who did not receive prophylactic eculizumab? A rate of 20-27% of death in two groups of patients is unusually high and hardly explained by a potential aHUS recurrence.

We considered a broad range of follow-up, and it is possible that the mortality was not exclusively related to aHUS recurrence. We included a one-year death in addition to the all-time follow-up death. 

One-year death rate was respectively in no eculizumab treatment, prophylactic and eculizumab treatment: 9%, 0% and 12%. We also included a supplementary Table with all causes and time of death (S4 Table)

Group Mortality Cause Time Post-transplant

Eculizumab treatment Hemorrhagic shock in the postoperative period 9 months

Eculizumab treatment Septic shock after hysterectomy and enterectomy 32 months

Eculizumab treatment Septic shock secondary to an infected foot ulcer 10 months

Eculizumab treatment Septic shock to urinary infection. 21 months

No Eculizumab use TMA recurrence 21 months

No Eculizumab use TMA recurrence 6 months

d) The discussion is more focused on the accessibility to eculizumab in Brazil rather than on the results of the study. A cost-effective use of eculizumab in renal transplantation is based in good part on genetic analysis, and this point needs to be more developed (see reference Zuber et al. JASN).

We agree and discuss the cost-effective use of eculizumab in renal transplantation based on genetic analysis:

The cost-effectiveness of accessing eculizumab for kidney transplant patients should be based on genetic analysis, thus, it is necessary to include the genetic profile for patients with suspicion for aHUS. In those classified at high to moderate risk according to KDIGO 2017 we strongly consider prophylaxis. As the opposite, we consider discontinuing the use in those without identified genetic variants, as well as the saving of 32 million euros realized by the STOPECU study (Fakhouri et al, 2021) after the discontinuation of 55 patients with an average of 24 months of follow-up.

Reviewer #2: This was a multicenter retrospective cohort study on 38 patients with ESKD due to aHUS who underwent kidney transplant. Patients were stratified into three groups: no use of eculizumab (No Eculizumab Group), use of eculizumab for treatment of after transplantation TMA (Therapeutic Group), and use of eculizumab for prophylaxis of aHUS recurrence (Prophylatic Group). In the No Eculizumab Group (n=11), there was a 91% graft loss due to the TMA. The hazard ratio of TMA graft loss was 0.07 [0.01-0.55], p = 0.012 in the eculizumab Prophylatic Group and 0.04 [0.00 - 0.28], p =0.002 in the eculizumab Therapeutic Group. The authors conclude that TMA graft loss in the absence of a specific complement-inhibitor was higher among the Brazilian cohort of kidney transplant patients. This finding reinforces the need of eculizumab use for treatment of aHUS kidney transplant patients.

This is a nice retrospective study in a group of patients with an ultra-rare disease.

Comments:

1. How many patients lost their graph in the Prophylactic and Treatment group? The manuscript just gives %

We provided the number in the text and Table 03.

Group Lost-Graft

Eculizumab treatment (n=17) TMA recurrence (n=1)

Prophylactic eculizumab (n=10) TMA recurrence (n=1)

No Eculizumab use (n=11) TMA recurrence (n=10) 

2. Please clarify: “42% underwent plasmapheresis and 79% received eculizumab”. This is not clear for total number of patients that received eculizumab was 27/38 = 71%

That's correct, 71% received eculizumab. We will adjust it.

3. “TMA graft loss occurred in 32% of cases and 17% died” please give actual numbers.

We provided the numbers through the text

4. Please provide details of the genetic analysis for the group treated with Eculizumab versus the group that was untreated.

We will send it in supplementary (S3 Table)

5. Table 2. 18 + 19 = 37. There is 1 patient missing.

Sorry, it was a mistake. There are 19 without testing and 19 with genetic variant analysis

6. Please provide details of the Eculizumab regime used. Was Eculizumab ever discontinued post-transplant? What was the criteria used? What was the outcome in patients that discontinued Eculizumab?

No patient was discontinued eculizumab after starting treatment. In the prophylactic group one patient evolved to graft loss due to TMA after hospitalization due to arteriovenous fistula thrombosis and had a delay of eculizumab infusion at that time. In the treatment group one patient lost the graft, probably due to a delay in starting eculizumab treatment, which was performed more than 120 days after diagnosis.

We provided these informations in the text

7. There are a few typos/grammar mistakes that need to be fixed.

We are sorry, we revised the manuscript for grammar mistakes

Reviewer #3: This interesting article highlights the efficacy of complement blockage in preventing and treating the aHUS recurrence after kidney transplantation. The results are consistent with those previously published by Zuber and Siedlecky, as you mentioned in the paper. It would had been very interesting if the genetic study of all patients was available. In spite of that, is important to remark that the patients with CFI mutations had TMA recurrence only if they have combined mutations

- P26 line 8:

"However, the inclusion of the eculizumab as a therapeutic option after transplantation has been significantly improving graft survival by preventing the recurrence of TMA"

In this line, only prophylactic eculizumab prevents recurrence. Please, clarify.

We modified to:

"However, the inclusion of the eculizumab as a therapeutic or prophylactic option after transplantation has been significantly improving graft survival by preventing the recurrence of TMA"

- In the discussion section, you have mentioned the probable causes associated to the high rate of acute rejection. However, you haven't showed data related to this assumption in the results section. Could you give more information?

Unfortunately, we could not retrieve more information about the episodes of acute rejection especially in the cases related to the untreated group (the majority were not biopsy proven). We add this information in the limitations section. 

- The differences in the survival probabilities between groups are not statistically significant, probably due to the small sample size. Please, remark this finding

We highlight these finding

Reviewer #4: I reviewed the manuscript PONE-D-21-24463 titled "Thrombotic microangiopathy after kidney transplantation: analysis of the Brazilian Atypical Hemolytic Uremic Syndrome cohort". This is a retrospective multicentris small cohort study that included only 38 aHUS KTx.

Because the results are confirmatory in nature the paper deserves not more than a Letter to the Editor.

There are 118 kidney transplant centers across Brazil but only ..... 6 accepted to include their patients in this retrospective study that took place between January 2007 and December 2019. during that period of 12 years how many patients were kidney transplanted across Brazil and how many in these 6 centers?

According to data from the Brazilian Transplant Association (ABTO) we had about 118 kidney transplant centers in Brazil. However, the majority of these transplant centers did not perform transplants in patients with aHUS or did not use eculizumab treatment. The access to eculizumab in Brazil was through compassionate use or judicialization that may be restricted to a small number of transplant centers. These reduced number of centers (n=6) received aHUS patients referred to other transplant programs to perform these special cases of transplants.

Were donor-specific alloantibodies (DSAs) looked for at the time of TMA recurrence, i.e., in many patients the cause of ESRD was not aHUS?

All patients had negative donor-specific alloantibodies (DSAs) at the time of TMA diagnosis as part of the aHUS work-up. The undetermined end-stage renal disease is likely to be attributed to patients who probably already have aHUS, but no initial diagnosis was made.

Across the 3 groups what were the causes of allograft loss?

The patients lost the grafts by TMA recurrence.

Group Lost-Graft

Eculizumab treatment (n=17) TMA recurrence (n=1)

Prophylactic eculizumab (n=10) TMA recurrence (n=1)

No Eculizumab use (n=11) TMA recurrence (n=10)

---

## [Decision Letter · Decision Letter 1]

24 Sep 2021

Thrombotic microangiopathy after kidney transplantation: analysis of the Brazilian Atypical Hemolytic Uremic Syndrome cohort

PONE-D-21-24463R1

Dear Dr. de Andrade,

We’re pleased to inform you that your manuscript has been judged scientifically suitable for publication and will be formally accepted for publication once it meets all outstanding technical requirements.

The revised manuscript is definitely improved. The authors have appropriately addressed all the reviewers critiques/comments.

Kind regards,

Giuseppe Remuzzi

Academic Editor

PLOS ONE

Additional Editor Comments (optional):

Reviewers' comments:

Reviewer's Responses to Questions

**Comments to the Author**

1. If the authors have adequately addressed your comments raised in a previous round of review and you feel that this manuscript is now acceptable for publication, you may indicate that here to bypass the “Comments to the Author” section, enter your conflict of interest statement in the “Confidential to Editor” section, and submit your "Accept" recommendation.

Reviewer #1: All comments have been addressed

Reviewer #2: All comments have been addressed

Reviewer #3: (No Response)

Reviewer #4: All comments have been addressed

2. Is the manuscript technically sound, and do the data support the conclusions?

Reviewer #1: Partly

Reviewer #2: Yes

Reviewer #3: (No Response)

Reviewer #4: Yes

3. Has the statistical analysis been performed appropriately and rigorously? 

Reviewer #1: I Don't Know

Reviewer #2: Yes

Reviewer #3: (No Response)

Reviewer #4: Yes

4. Have the authors made all data underlying the findings in their manuscript fully available?

Reviewer #1: Yes

Reviewer #2: Yes

Reviewer #3: (No Response)

Reviewer #4: Yes

5. Is the manuscript presented in an intelligible fashion and written in standard English?

Reviewer #1: Yes

Reviewer #2: Yes

Reviewer #3: (No Response)

Reviewer #4: Yes

6. Review Comments to the Author

Reviewer #1: (No Response)

Reviewer #2: Authors have addressed my comments appropriately. I have no further comments.

Reviewer #3: The authors have commented and corrected all the comments made by this reviewer.

Eventhough the results of the study are in line with the previously published, as aHUS is an ultra-rare disease I think it is convenient to publish this article in order to increase the knowledge of this entity

Reviewer #4: The authors have responded to my questions/concerns. I have no additional comment. However, I think that the paper could be condensed as a Letter to the Editor.

7. PLOS authors have the option to publish the peer review history of their article (what does this mean?). If published, this will include your full peer review and any attached files.

Reviewer #1: No

Reviewer #2: **Yes: **Fernando C Fervenza, MD

Reviewer #3: **Yes: **Ana Ávila

Reviewer #4: No

---

## [Editor Report · Acceptance letter]

21 Oct 2021

PONE-D-21-24463R1 

Thrombotic microangiopathy after kidney transplantation: analysis of the Brazilian Atypical Hemolytic Uremic Syndrome cohort 

Dear Dr. de Andrade:

I'm pleased to inform you that your manuscript has been deemed suitable for publication in PLOS ONE. Congratulations! Your manuscript is now with our production department. 

Kind regards, 

on behalf of

Prof. Giuseppe Remuzzi 

Academic Editor

PLOS ONE